# Large-scale L-BFGS using MapReduce

**Weizhu Chen, Zhenghao Wang, Jingren Zhou**
Microsoft
{wzchen,zhwang,jrzhou}@microsoft.com

## Abstract

L-BFGS has been applied as an effective parameter estimation method for various machine learning algorithms since 1980s. With an increasing demand to deal with massive instances and variables, it is important to scale up and parallelize L-BFGS effectively in a distributed system. In this paper, we study the problem of parallelizing the L-BFGS algorithm in large clusters of tens of thousands of shared-nothing commodity machines. First, we show that a naive implementation of L-BFGS using Map-Reduce requires either a significant amount of memory or a large number of map-reduce steps with negative performance impact. Second, we propose a new L-BFGS algorithm, called Vector-free L-BFGS, which avoids the expensive dot product operations in the two loop recursion and greatly improves computation efficiency with a great degree of parallelism. The algorithm scales very well and enables a variety of machine learning algorithms to handle a massive number of variables over large datasets. We prove the mathematical equivalence of the new Vector-free L-BFGS and demonstrate its excellent performance and scalability using real-world machine learning problems with billions of variables in production clusters.

## 1 Introduction

In the big data era, many applications require solving optimization problems with billions of variables on a huge amount of training data. Problems of this scale are more common nowadays, such as Ads CTR prediction[1] and deep neural network[2]. The other trend is the wide adoption of map-reduce [3] environments built with commodity hardware. Those large-scale optimization problems are often expected to be solved in a map-reduce environment where big data are stored.

When a problem is with huge number of variables, it can be solved efficiently only if the storage and computation cost are maintained effectively. Among a diverse collection of large-scale optimization methods, Limited-memory BFGS (L-BFGS)[4] is one of the frequently used optimization methods in practice[5]. In this paper, we study the L-BFGS implementation for billion-variable scale problems in a map-reduce environment. The original L-BFGS algorithm and its update procedure were proposed in 1980s. A lot of popular optimization software packages implement it as a fundamental building block. Approaches to apply it in a problem with up to millions of variables are well studied and implemented in various optimization packages [6]. However, studies about how to scale L-BFGS into billions of variables are still in their very early stages. For such a massive scale, the parameters, their gradients, and the associated L-BFGS historical states are not only too large to be stored in the memory of a single computation node, but also create too huge computation complexity for a processor or multicores to conquer it within reasonable time. Therefore, it is critical to explore an effective decomposition over both examples and models via distributed learning. Yet, to our knowledge, there is still very limited work to explore billion-variable scale L-BFGS. This directly leads to the consequence that very little work can scale various machine learning algorithms up to billion-variable scale using L-BFGS on map-reduce.

In this paper, we start by carefully studying the implementation of L-BFGS in map-reduce environment. We examine two typical L-BFGS implementations in map-reduce and present their scaling obstacles. Particularly, given a problem with $d$ variables and $m$ historical states to approximate Hessian [5], traditional implementation[6][5], either need to store $2md$ variables in memory or need to perform $2m$ map-reduce steps per iteration. This clearly creates huge overhead for the problem with billions of variables and prevents a scalable implementation in map-reduce.

To conquer these limitations, we reexamine the original L-BFGS algorithm and propose a new L-BFGS update procedure, called Vector-free L-BFGS (VL-BFGS), which is specifically devised for distributed learning with huge number of variables. In particular, we replace the original L-BFGS update procedure depending on vector operations, as known as two-loop recursion, by a new procedure only relying on scalar operations. The new two-loop recursion in VL-BFGS is mathematically equivalent to the original algorithm but independent on the number of variable. Meanwhile, it reduces the memory requirement from $O(md)$ to $O(m^2)$ where $d$ could be billion-scale but $m$ is often less than 10. Alternatively, it only require 3 map-reduce steps compared to $2m$ map-reduce steps in another naive implementation.

This new algorithm enables the implementation of a collection of machine learning algorithms to scale to billion variables in a map-reduce environment. We demonstrate its scalability and advantage over other approaches designed for large scale problems with billions of variables, and share our experience after deploying it into an industrial cluster with tens of thousands of machines.

## 2  Related Work

L-BFGS [4][7] is a quasi-newton method based on the BFGS [8][9] update procedure, while maintaining a compact approximation of Hessian with modest storage requirement. Traditional implementation of L-BFGS follows [6] or [5] using the compact two-loop recursion update procedure. Although it has been applied in the industry to solve various optimization problems for decades, recent work, such as [10][11], continue to demonstrate its reliability and effectiveness over other optimization methods. In contrast to our work, theirs implemented L-BFGS on a single machine while we focus on the L-BFGS implementation in a distributed environment.

In the context of distributed learning, there recently have been extensive research break-through. GraphLab [12] built a parallel distributed framework for graph computation. [13] introduced a framework to parallelize various machine learning algorithms in a multi-core environment. [14] applied the ADMM technique into distributed learning. [15] proposed a delayed version of distributed online learning. General distributed learning techniques closer to our work are the approaches based on parallel gradient calculation followed by a centralized algorithm ([7][16][17]). Different from our work, theirs built on fully connected environment such as MPI while we focus on the map-reduce environment with loose connection. Their centralized algorithm is often the bottleneck of the whole procedure and limits the scalability of the algorithm. For example, [17] clearly stated that it is impractical for their L-BFGS algorithm to run their large dataset due to huge memory consumption in the centralized algorithm although L-BFGS has been shown to be an excellent candidate for their problem. Moreover, the closest to our work lies in applying L-BFGS in the map-reduce-like environment, such as [18][2]. They are solving large-scale problems in a map-reduce adapted environment using L-BFGS. [18] run L-BFGS on a map-reduce plus AllReduce environment to demonstrate the power of large-scale learning with map-reduce. Although it has been shown to scale up to billion of data instances with trillion entries in their data matrix, the number of variables in their problem is only about 16 million due to the constraints in centralized computation of L-BFGS direction. [2] used L-BFGS to solve the deep learning problem. It introduced the parameter servers to split a global model into multiple partitions and store each partition separately. Despite their successes, from the algorithmic point of view, their two-loop recursion update procedure is still highly dependent on the number of variable. Compared with these work, our proposed two-loop recursion updating procedure is independent on the number of variables and with much better parallelism. Furthermore, the proposed algorithm can run on pure map-reduce environment while previous work [2] and [18] require special components such as AllReduce or parameter servers. In addition, it is straightforward for previous work, such as [2][18][17], to leverage our proposal to scale up their problem into another order of magnitude in terms of number of variables.

# 3 L-BFGS Algorithm

Given an optimization problem with $d$ variables, BFGS requires to store a dense $d$ by $d$ matrix to approximate the inverse Hessian, where L-BFGS only need to store a few vectors of length $d$ to approximate the Hessian implicitly. Let us denote $f$ as the objective function, $g$ as the gradient and $\cdot$ as the dot product between two vectors. L-BFGS maintains the historical states of previous $m$ (generally $m = 10$) updates of current position $x$ and its gradient $g = \nabla f(x)$.

In L-BFGS algorithm, the historical states are represented as the last $m$ updates of form $s_k = x_{k+1} - x_k$ and $y_k = g_{k+1} - g_k$ where $s_k$ represents the position difference and $y_k$ represents the gradient difference in iteration $k$. Each of them is a vector of length $d$. All of these $2m$ vector with the original gradient $g_k$ will be used to calculate a new direction in line 3 of Algorithm 1.

---

**Algorithm 1:** L-BFGS Algorithm Outline

    **Input**: starting point $x_0$, integer history size $m > 0$, k=1;
    **Output**: the position $x$ with a minimal objective function
**1** **while** *no converge* **do**
**2**      Calculate gradient $\nabla f(x_k)$ at position $x_k$ ;
**3**      Compute direction $p_k$ using Algorithm 2 ;
**4**      Compute $x_{k+1} = x_k + \alpha_k p_k$ where $\alpha_k$ is chosen to satisfy Wolfe conditions;
**5**      **if** $k > m$ **then**
**6**         Discard vector pair $s_{k-m}, y_{k-m}$ from memory storage;;
**7**      **end**
**8**      Update $s_k = x_{k+1} - x_k, y_k = \nabla f(x_{k+1}) - \nabla f(x_k), k = k + 1$ ;
**9** **end**

---

**Algorithm 2:** L-BFGS two-loop recursion

    **Input**: $\nabla f(x_k)$, $s_i, y_i$ where $i = k - m, ..., k - 1$
    **Output**: new direction $p$
**1** $p = -\nabla f(x_k)$ ;
**2** **for** $i \leftarrow k - 1$ **to** $k - m$ **do**
**3**      $\alpha_i \leftarrow \frac{s_i \cdot p}{s_i \cdot y_i}$ ;
**4**      $p = p - \alpha_i \cdot y_i$ ;
**5** **end**
**6** $p = (\frac{s_{k-1} \cdot y_{k-1}}{y_{k-1} \cdot y_{k-1}})p$
**7** **for** $i \leftarrow k - m$ **to** $k - 1$ **do**
**8**      $\beta = \frac{y_i \cdot p}{s_i \cdot y_i}$ ;
**9**      $p = p + (\alpha_i - \beta) \cdot s_i$;
**10** **end**

---

The core update procedure in Algorithm 1 is the line 3 to calculate a new direction $p_k$ using $s$ and $y$ with current gradient $\nabla f(x_k)$. The most common approach for this calculation is the two-loop recursion in Algorithm 2[5][6]. It initializes the direction $p$ with gradient and continues to update it using historical states $y$ and $s$. More information about two-loop recursion could be found from [5].

# 4 A Map-Reduce Implementation

The main procedure in Algorithm 1 lies in Line 2, 3 and 4. The calculation of gradient in Line 2 can be straightforwardly parallelized by dividing the data into multiple partitions. In the map-reduce environment, we can use one map step to calculate the partial gradient for partial data and one reduce to aggregate them into a global gradient vector. The verification of the Wolfe condition only depends on the calculation of the objective function following the line search procedure[5]. So thus Line 4 can also be easily parallelized following the same approach as Line 2. Therefore, the challenge in the L-BFGS algorithm is Line 3. In other words,the difficulties come from the calculation of the two-loop recursion, as shown in Algorithm 2.

## 4.1 Centralized Update

The simplest implementation for Algorithm 2 may be to run it in a single processor. We can easily perform this in a singleton reduce. However, the challenge is that Algorithm 2 requires $2m + 1$ vectors and each of them has a length of $d$. This could be feasible when $d$ is in million scale. Nevertheless, when $d$ is in billion scale, either the storage or the computation cost becomes a significant challenge and makes it impractical to implement it in map-reduce. Given the Ads CTR prediction task [1] as an example, there are more than 1 billion of features. If we set $m = 10$ in a linear model, it will produce $21 * 1 = 21$ billion variables. Even if we compactly use a single-precision floating point to represent a variable, it requires $84$ GB memory to store the historical states and gradient. For a map-reduce cluster built from commodity hardware and shared with other applications, this is generally unfeasible nowadays. For example, for the cluster into which we deployed the L-BFGS, its maximal memory limitation for a map-reduce step is 6 GB.

## 4.2 Distributed Update

Due to the storage limitation in centralized update, an alternative is to store $s$ and $y$ into multiple partitions without overlap and use a map-reduce step to calculate every dot product, such as $s_i \cdot p$ and $s_i \cdot y_i$ in Line 3 of Algorithm 2. Yet, if each dot product within the for-loop in Algorithm 2 requires a map-reduce step to perform the calculation, this will result in at least $2m$ map-reduce steps in a two-loop recursion. If we call Algorithm 2 for $N$ times(iterations) in Algorithm 1, it will lead to $2mN$ map-reduce steps. For example, if $m = 10$ and $N = 100$, this will produce 2000 map-reduce steps in a map-reduce job. Unfortunately, each map-reduce step will bring significant overhead due to the scheduling cost and application launching cost. For a job with thousands of map-reduce steps, both these cost often dominate the overall running time and make the useful computational time spent in algorithmic vector operations negligible. Moreover, given our current production cluster as an example, a job with such a huge number of map-reduce step is too large for execution. It will trigger a compilation timeout error before becoming too complicated for an execution engine to execute it.

## 5 Vector-free L-BFGS

For the reasons mentioned, a feasible two-loop recursion procedure has to limit both the memory consumption and the number of map-reduce steps per iteration. To strictly limit the memory consumption in Algorithm 2, we can not store the $2m + 1$ vectors with length $d$ in memory unless $d$ is only up to million scale. To comply with the allowable map-reduce steps per iteration, it is neither practical to perform map-reduce steps within the for-loop in Algorithm 2. Both of these assumptions motivate us to carefully re-examine Algorithm 2 and lead to the proposed algorithm in this section.

### 5.1 Basic Idea

Before illustrating the new procedure, let us describe following three observations in Algorithm 2 that guide the design of the new procedure in Algorithm 3:

1. All inputs are invariable during Algorithm 2.
2. All operations applied on $p$ are linear with respect to the inputs. In other words, $p$ could be formalized as a linear combination of the inputs although its coefficients are unknown.
3. The core numeric operation is the dot product between two vectors.

Observation 1 and 2 motivate us to formalize the inputs as $(2m + 1)$ invariable base vectors.

$$b_1 = s_{k-m}, b_2 = s_{k-m+1}, ..., b_m = s_{k-1} \tag{1}$$

$$b_{m+1} = y_{k-m}, b_{m+2} = y_{k-m+1}, ..., b_{2m} = y_{k-1} \tag{2}$$

$$b_{2m+1} = \nabla f(x_i) \tag{3}$$

So thus we can represent $p$ as a linear combination of $b_i$. Assume $\delta$ as the scalar coefficients in this linear combination, we can write $p$ as:

$$p = \sum_{k=1}^{2m+1} \delta_k b_k \tag{4}$$

Since $b_k$ are the inputs and invariants during the two-loop recursion, if we can calculate the coefficients $\delta_k$, we can proceed to calculate the direction $p$.

Following observation 3 with an re-examination of Algorithm 2, we classify the dot product operations into two categories in terms of whether $p$ is involved in the calculation. For the first category only involving the dot product between the inputs $(s_i, y_i)$, a straightforward intuition is to pre-compute their dot products to produce a scalar, so as to replace each dot product with a scalar in the two-loop recursion. However, the second category of dot products involving $p$ can not follow this same procedure. Because the direction $p$ is ever-changing during the for loop, any dot products involving $p$ can not be settled or pre-computed. Fortunately, thanks to the linear decomposition of $p$ in observation 2 and Eqn.4, we can decompose any dot product involving $p$ into a summation of dot products with its based vectors and corresponding coefficients. This new elegant mathematical procedure only happens after we formalize $p$ as the linear combination of the base vectors.

## 5.2 The VL-BFGS Algorithm

We present the algorithmic procedure in Algorithm 3. Let us denote the results of dot products between every two base vectors as a scalar matrix of $(2m + 1) * (2m + 1)$ scalars. The proposed VL-BFGS algorithm only takes it as the input. Similar as the original L-BFGS algorithm, it has a two-loop recursion, but all the operations are only dependent on scalar operations. In Line 1-2, it assigns the initial values for $\delta_i$. This is equivalent to Line 1 in Algorithm 2 to use opposite direction of gradient as the initial direction. The original calculation of $\alpha_i$ in Line 6 relies on the direction vector $p$. It is worth noting that $p$ is variable within the first loop in which $\delta$ is updated. So thus we can not pre-compute any dot product involving $p$. However, as mentioned earlier and according to observation 2 and Eqn.4, we can formalize $b_j \cdot p$ as a summation from a list of dot products between base vectors and corresponding coefficients, as shown in Line 6 of Algorithm 3. Meanwhile, since all base vectors are invariable, their dot products can be pre-computed and replaced with scalars, which then multiply the ever-changing $\delta_l$. But these are only scalar operations and they are extremely efficient. Line 7 continues to update scalar coefficient $\delta_{m+j}$, which is equivalent to update the direction $p$ with respect to the base vector $b_{m+j}$ or corresponding $y_j$. This whole procedure is the same when we apply it to Line 14 and 15. With the new formalization of $p$ in Eqn.4 and the

---

**Algorithm 3:** Vector-free L-BFGS two-loop recursion

    **Input**: $(2m + 1) * (2m + 1)$ dot product matrix between $b_i$
    **Output**: The coefficients $\delta_i$ where $i = 1, 2, ...2m + 1$
**1**  **for** $i \leftarrow 1$ **to** $2m + 1$ **do**
**2**    |  $\delta_i = i \leq 2m\ ?\ 0 : -1$
**3**  **end**
**4**  **for** $i = k - 1$ **to** $k - m$ **do**
**5**    |  $j = i - (k - m) + 1$ ;
**6**    |  $\alpha_i \leftarrow \frac{s_i \cdot p}{s_i \cdot y_i} = \frac{b_j \cdot p}{b_j \cdot b_{m+j}} = \frac{\sum_{l=1}^{2m+1} \delta_l b_l \cdot b_j}{b_j \cdot b_{m+j}}$ ;
**7**    |  $\delta_{m+j} = \delta_{m+j} - \alpha_i$ ;
**8**  **end**
**9**  **for** $i \leftarrow 1$ **to** $2m + 1$ **do**
**10**   |  $\delta_i = (\frac{b_m \cdot b_{2m}}{b_{2m} \cdot b_{2m}})\delta_i$
**11** **end**
**12** **for** $i \leftarrow k - m$ **to** $k - 1$ **do**
**13**   |  $j = i - (k - m) + 1$ ;
**14**   |  $\beta = \frac{b_{m+j} \cdot p}{b_j \cdot b_{m+j}} = \frac{\sum_{l=1}^{2m+1} \delta_l b_{m+j} \cdot b_l}{b_j \cdot b_{m+j}}$ ;
**15**   |  $\delta_j = \delta_j + (\alpha_i - \beta)$
**16** **end**

---

invariability of $y_i$ and $s_i$ during Algorithm 2, Line 4 in Algorithm 2 updating with $y_i$ (equivalent to $b_{m+j}$) is mathematically equivalent to Line 7 in Algorithm 3, so as Line 9 in Algorithm 2 and Line 15 in Algorithm 3. For other lines between these two algorithms, it is easy to infer their equivalence with the consideration of Eqn.1-4. Thus, Algorithm 3 is mathematically equivalent to Algorithm 2.

### 5.3 Complexity Analysis and Comparison

Using the dot product matrix of scalars as the input, the calculation in Algorithm 3 is substantially efficient, since all the calculation is based on scalars. Altogether, it only requires $8m^2$ multiplications between scalars in the two for-loops. This is tiny compared to any vector operation involving billion-scale of variables. Thus, it is not necessary to parallelize Algorithm 3 in implementation.

To integrate Algorithm 3 as the core step in Algorithm 1, there are two extra steps we need to perform before and after it. One is to calculate the dot product matrix between the $(2m + 1)$ base vectors. Because all base vectors have the same dimension $d$, we can partition them using the same way and use one map-reduce step to calculate the dot product matrix. This computation is greatly parallelizable and intrinsically suitable for map-reduce. Even without the consideration of parallization, a first glance tells us it may require about $4m^2$ dot products. However, since all the $s_i$ and $y_i$ except the first ones are unchanged in a new iteration, we can save the tiny dot product matrix and reuse most entries across iterations. With the consideration of the commutative law of multiplication since $s_i \cdot y_j \equiv y_j \cdot s_i$, each new iteration only need to calculate $6m$ new dot products which involve new $s_k$, $y_k$ and $g_k$. Thus, the complexity is only $6md$ and this calculation is fully parallel in map-reduce, with each partition only calculating a small portion of $6md$ multiplications.

The other and the final step is to calculate the new direction $p$ based on $\delta_i$ and the base vectors. The complexity is another $2md$ multiplications, which means the overall complexity of the algorithm is $8md$ multiplications. Since the overall $\delta$ is just a tiny vector with $2m + 1$ dimensions, we can join it with all the other base vectors, and then use the same approach as dot product calculation to produce the final direction $p$ using Eqn.4. A single map-reduce step is sufficient for this final step. Altogether, without considering the gradient calculation which is same to all algorithms, VL-BFGS only require 3 map-reduce steps for one iteration in the update.

For the centralized update approach in section 4.1, it also requires $6md$ multiplications in each two loop recursion. In addition to being a centralized approach, as we analyzed above, it requires $(2m + 1) * d$ memory storage. This clearly limits its applications to large-scale problems. On the other hand, VL-BFGS in Algorithm 3 only requires $(2m+1)^2$ memory storage and is independent on $d$. For the distributed approach in section 4.2, it requires at least $2m$ map-reduce step in a two-loop recursion. Given the number of iteration as $N$ (generally $N > 100$), the total number of map-reduce steps is $2mN$. Fortunately, the VL-BFGS only requires $3N$ map-reduce steps. In summary, VL-BFGS algorithm enjoys a similar overall complexity but it is born with massive degree of parallelism. For problem with billion scale of variables, it is the only map-reduce friendly implementation of the three different approaches.

## 6 Experiment and Discussion

As demonstrated above, it is clear that VL-BFGS has a better scalability property than original L-BFGS. Although it is always desirable to invent an exact algorithm that could be mathematically proved to obtain a better scalability property, it is beneficial to demonstrate the value of larger number of variables with an industrial application. On the other hand, for a problem with billions of variables, there are existing practical approaches to reduce it into a smaller number of variables and then solve it with traditional approaches designed for centralized algorithm. In this section, we justify the value of learning large scale variables and simultaneously compare it with the hashing approach, and finally demonstrate the scalability advantage of VL-BFGS.

### 6.1 Dataset and Experimental Setting

The dataset we used is from an Ads Click-through Rate (CTR) prediction problem [1] collected from an industrial search engine. The click event (click or not) is used as the label for each instance. The features include the terms from a query and an Ad keyword along with the contextual information such as Ad position, session-related information and time. We collect 30 days of data and split them into training and test set chronologically. The data from the first 20 days are used as the training set and rest 10 days are used as test set. The total training data have about 12 billions instances and another 6 billion in testing data. There are 1,038,934,683 features the number of non-zero features per instance is about 100 on average. Altogether it has about 2 trillion entries in the data matrix.

Table 1: Relative AUC Performance over different number of variables

| K | Relative AUC Performance |
| --- | --- |
| Baseline(K=1,038,934,683) | 0.0% |
| K=250 millions | -0.1007388% |
| K=100 millions | -0.1902843% |
| K= 10 millions | -0.3134094% |
| K= 1 millions | -0.5701142% |

Table 2: Relative AUC Performance over different number of Hash bits

| K | Relative AUC Performance |
| --- | --- |
| Baseline(K=1,038,934,683) | 0.0% |
| K=64 millions(26 bits) | -0.1063033% |
| K=16 millions(24 bits) | -0.2323647% |
| K= 4 millions(22 bits) | -0.3300788% |
| K= 1 millions(20 bits) | -0.5080904% |

We run logistic regression training, so thus each feature corresponds to a variable. The model is evaluated based on the testing data using Area Under ROC Curve [19], denoted as AUC. We set the historical state length $m = 10$ and enforce L1[20] regularizer to avoid overfitting and achieve sparsity. The regularizer parameter is tuned following the approach in [18].

We run the experiment in a shared cluster with tens of thousands of machines. Each machine has up to 12 concurrent vertices. A vertex is generally a map or reduce step with an allocation of 2 cores and 6G memory. There are more than 1000 different jobs running simultaneously but this number also varies significantly. We split the training data into 400 partitions and allocate 400 tokens for this job, which means this job can use up to 400 vertices at the same time. When we partition vectors to calculate their dot products, our strategy is to allocate up to 5 million entries in a partial vector. For example, 1 billion variables will be split into 200 partitions evenly.

We use the model trained with original 1 billion features as the baseline. All the other experiments are compared with it. Since we are not allowed to exhibit the exact AUC number due to privacy consideration, we report the relative change compared with the baseline. The scale of the dataset makes any relative AUC change over 0.001% produce a p-value less than 0.01.

## 6.2 Value of Large Number of Variables

To reduce the number of variables in the original problem, we sort the features based on their frequency in the training data. If we plan to reduce the problem to $K$ variables, we keep the top $K$ frequent features. The baseline without filtering is equivalent to $K = 1,038,934,683$. We choose different $K$ values and report the relative AUC number in Table 1.

The table shows that while we reduce the number of variables, the results consistently decline significantly. When the number of variables is 1 million, the drop is more than 0.5% . This is considerably significant for the problem. Even when we increase the number of variable up to 250 million, the decline is still obvious and significant. This demonstrates that the large number of variables is really needed to learn a good model and the value of learning with billion-scale of variables.

## 6.3 Comparison with Hashing

We follow the approach in [21][18] to calculate a new hash value for each original feature value based on a hash function in [18]. The number of hash bits ranges from 20 to 26. Experimental results compared with the baseline in terms of relative AUC performance are presented in Table 2

Consistently with previous results, all the hashing experiments result in degradation. For the experiment with 20 bits, the degradation is 0.5%. This is a substantial decline for this problem. When we increase the number of bits till 26, the gap becomes smaller but still noticeable. All of these consis-

tently demonstrate that the hashing approach will sacrifice noticeable performance. It is beneficial to train with large-scale number of raw features.

## 6.4 Training Time Comparison

We compare the L-BFGS in section 4.1 with the proposed VL-BFGS. To enable a larger number of variable support for L-BFGS, we reduce the $m$ parameter into 3. We conduct the experiments with varying number of feature number and report their corresponding running time. We use the original data after hashing into 1M features as the baseline and compare all the other experiments with it and report the relative training time for same number of iterations. We run each experiment 5 times and report their mean to cope with the variance in each run. The results with respect to different hash bits range from 20 to 29 and the original 1B features are shown in figure 1. When the number of features is less than 10M, the original L-BFGS has a small advantage over VL-BFGS. However, when we continue to increase the feature number, the running time of L-BFGS grows quickly while that of VL-BFGS increases slowly. On the other hand, when we increase the feature number to 512M, the L-BFGS fails with an out-of-memory exception, while VL-BFGS can easily scale to 1B features. All of these clearly show the scalability advantage of VL-BFGS over traditional L-BFGS.

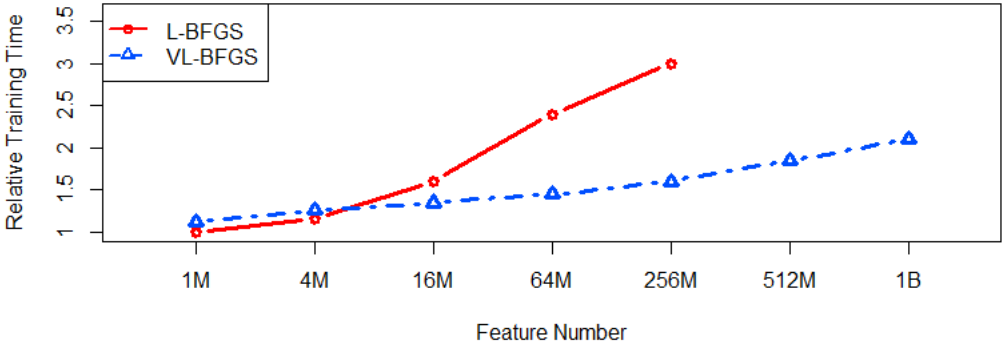

Figure 1: Training time over feature number.

## 7  Conclusion

We have presented a new vector-free exact L-BFGS updating procedure called VL-BFGS. As opposed to original L-BFGS algorithm in map-reduce, the core two-loop recursion in VL-BFGS is independent on the number of variables. This enables it to be easily parallelized in map-reduce and scale up to billions of variables. We present its mathematical equivalence to original L-BFGS, show its scalability advantage over traditional L-BFGS in map-reduce with a great degree of parallelism, and perform experiments to demonstrate the value of large-scale learning with billions of variables using VL-BFGS. Although we emphasis the implementation on map-reduce in this paper, VL-BFGS can be straightforwardly utilized by other distributed frameworks to avoid their centralized problem and scale up their algorithms. In short, VL-BFGS is highly beneficial for machine learning algorithms relying on L-BFGS to scale up to another order of magnitude.

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
