[Reviews · NeurIPS 2014]

Submitted by Assigned_Reviewer_7

This paper discusses large-scale L-BFGS by using MapReduce. The main contribution is that it proposed a new algorithm for finding the search direction. This new algorithms requires few MapReduce operations.

Comments:

- The calculation of the search direction is by a sequential for loop. The authors have a clever way to make it suitable for MapReduce. The operations become more complicated, but the communication cost is reduced.

- I think it's better if the authors can provide running time comparison.

Summary: The main contribution is that it proposed a new algorithm for finding the search direction of LBFGS. The technique can effectively parallelize the calculation of the direction.

Submitted by Assigned_Reviewer_31

This paper proposes a new vector free updating formulation for the two loop recursion in L-BFGS. The main insight is that the gradient direction can be expressed as a linear combination of historical state vectors. As a result, the dot product between the state vector and the gradient can be computed as a linear combination of dot products between state vectors which can be precomputed and independent of the size of input vectors. This results in a scalable and efficient implementation of L-BFGS in the MapReduce framework. The proposed method is evaluated on a real-world large scale problem with billions of variables thus demonstrating its efficiency. It's also compared to hashing scheme and shows better accuracy.

Overall, the proposed method is efficient and practical in large-scale settings. The paper is also well written.

There are some typos as follows.
- Line 82: "in contract to" -> "in contrast to"
- Line 322: "compare the it" -> "compare it"
Summary: The proposed method is efficient and practical for solving large-scale problems using L-BFGS especially with a large number of features. The paper is well written.

Submitted by Assigned_Reviewer_40

The idea is based on the fact that the search direction is a linear
combination of base vectors, instead of following l-BFGS two-loop
recursion, the search direction can be indirectly computed from the
linear combination coefficients, so that multiple rounds of
communication can be avoided. The experiments illustrate the
scalability.

The idea is simple and effective. The algorithm seems to be
correct. The presentation is clear.

Summary: This paper addresses the problem of applying l-BFGS algorithm on very
high dimemsion data in a distributed environment. The solution is
smart and the paper is well written.
Author Feedback
Author rebuttal: We’d like to thank all the reviewers for their valuable comments.

We will continue to improve the presentation, fix typos, and address comments in our revision. We also plan to enhance the experimental section to provide detailed runtime study and disclose more production system internal information in the final version.